# Impacts of the National Forest Rehabilitation Plan and Human-Induced Environmental Changes on the Carbon and Nitrogen Balances of the South Korean Forests

Hyung-Sub Kim [1,2], Florent Noulèkoun [3], Nam-Jin Noh [4] and Yo-Whan Son [2,*]

1. Institute of Life Science and Natural Resource, Korea University, Seoul 02841, Korea; khs9461@korea.ac.kr
2. Division of Environmental Science and Ecological Engineering, Korea University, Seoul 02841, Korea
3. Agroforestry Systems and Ecology Laboratory, Division of Environmental Science and Ecological Engineering, Korea University, Seoul 02841, Korea; florentnoulekoun@yahoo.fr
4. Department of Forest Resources, Kangwon National University, Chuncheon 24341, Korea; njnoh@kangwon.ac.kr
* Correspondence: yson@korea.ac.kr

**Abstract:** Humans have affected the carbon (C) and nitrogen (N) cycles in forests; however, the quantification of the responses of forest C and N balances to human activities is limited. In this study, we have quantified the impacts of the long-term national forest rehabilitation plan and the contribution of the increase in air temperature, $CO_2$ concentration, and N deposition on the C and N balances of the South Korean forests during 1973–2020 by using a biogeochemical model. During the simulation period, the C balance increased from 0.2 to 4.3 Mg C ha$^{-1}$ year$^{-1}$, and the N balance increased from 0.2 to 17.4 kg N ha$^{-1}$ year$^{-1}$. This resulted in the storage of 825 Tg C and 3.04 Tg N by the whole South Korean forests after the national forest rehabilitation plan. The increase in air temperature, $CO_2$ concentration, and N deposition contributed $-11.5$, 17.4, and 177 Tg C to the stored C stock, respectively, and $-25.4$, 8.90, and 1807 Mg N to the stored N stock, respectively. This study provides references for future forest rehabilitation efforts and broadens our knowledge on the impacts of human-induced environmental changes on the C and N balances of forests.

**Keywords:** modeling; human activities; climate change; air temperature; $CO_2$ concentration; nitrogen deposition

## 1. Introduction

Carbon (C) and nitrogen (N) balances in a forest determines the supporting role of the forest for human beings. The C balance, determined by the photosynthetic C gain of trees and respiratory C loss from trees and dead organic matter (DOM), plays a key role in determining the climate regulation of a forest (e.g., atmospheric $CO_2$ sequestration) [1]. The N balance, the difference between N input through N deposition and biological N fixation and N output through N leaching and denitrification, affects forest productivity, ground and surface water quality, and $N_2O$ emission [2,3]. Thus, quantifying the C and N balances has been the core focus since the beginning of forest science [4–6]. However, human activities have impacted C and N cycles, and failure to account for the impacts of human activities in quantifying the C and N balances may lead to unreliable estimates.

Humans have directly affected C and N balances in forests through deforestation and forest rehabilitation. These activities can result in eitfher the loss or accumulation of C and N, thereby changing the whole C and N cycles. For instance, the Intergovernmental Panel on Climate Change reported that the estimated global $CO_2$ emissions from land-use change averaged 0.5–2.7 Gt C year$^{-1}$ in the 1990s [7]. A global meta-analysis study showed that the mean changing rates of soil C and N stocks after afforestation were 0.5 Mg C ha$^{-1}$ year$^{-1}$ and $-0.01$ Mg N ha$^{-1}$ year$^{-1}$, respectively [8]. Moreover, humans have indirectly impacted the C and N balances of forests by increasing air temperature, $CO_2$ concentration, and

N deposition through the use of fossil fuel and N fertilizer [9–13]. For example, Free-Air $CO_2$ Enrichment experiments showed that increased $CO_2$ concentration enhanced forest productivity [14]. A measurement-based study in Europe reported that both ecosystem respiration and gross primary production increased with mean annual temperature at the site [15]. A literature review study on the impact of N deposition on soil N cycle concluded that the increased N deposition would adversely affect other N input pathways (e.g., biological N fixation) and increase N leaching in an "N-rich" forest [16].

Modeling allows exploration of the C and N balances responses to changing environmental factors at broad spatial and temporal scales [17]. For example, different biogeochemical models have been used to understand historical changes in the C fluxes in response to climate change [18], to quantify the turnover times of C and N at present time [19], and to anticipate the responses of the C and N balances to future global changes [20]. Moreover, modeling studies have contributed toward establishing governmental decisions, including forest management and greenhouse gas reduction [21]. In this context, modeling has been a core research framework in forest C and N balances studies.

South Korean forests are one of the best places to study the effects of human activities on the C and N balances because they have one of the most notable deforestation and forest rehabilitation histories. During the Japanese occupation of Korea (1910–1945), more than 63 million $m^3$ of timber was exploited to support Japan's construction, military, industrial, and other domestic needs [22]. The Korean War (1950–1953) devastated more than half of the forest land. This decreased the average stem volume to 10.5 $m^3$ $ha^{-1}$, which was approximately 36% of the prewar estimates [23]. However, the Korean government launched the national forest rehabilitation plan in 1973, which led to the successful restoration of the Korean forest [24]. Meanwhile, air temperature and $CO_2$ concentration increased at rates of 0.28 °C and 19.2 ppm per decade from 1973 to 2020, respectively [25]. Moreover, the emissions of $NO_x$ increased at a rate of 8.9 Kt N $year^{-1}$ from the 1970s to the early 2000s in South Korea [26,27].

The long forest rehabilitation history of South Korea has made available long-term data on forest growth and productivity and enabled several modeling studies for South Korean forests. For example, a forest C model estimated C dynamics from 1954 to 2012 [28]. A C budget model specified for South Korean forests quantified the impact of the national forest rehabilitation plan on the C cycle [29]. An ecosystem service model assessed the economic viability of the national forest rehabilitation plan [30]. However, these studies have three limitations. First, the effect of N availability on C balance was not considered [31]. Second, the impact of the national forest rehabilitation plan on N balance was not assessed, although N balance affects C sequestration capacity, water quality, and $N_2O$ emissions [2,3]. Third, the increases in the C balance of the South Korean forests were attributed to the national forest rehabilitation plan, without considering the contribution of environmental factors, such as air temperature, $CO_2$ concentration, and N deposition.

This study had two main objectives. The first objective was to estimate the C and N dynamics of the South Korean forests after the national forest rehabilitation plan considering the increase in air temperature, $CO_2$ concentration, and N deposition. A biogeochemical model simulating C and N dynamics, Forest Biomass and Dead organic Carbon and Nitrogen (FBD-CAN), was chosen for the estimation because of its proven reliability and adaptability to the South Korean forests [19,28,32]. The reliability of the model estimation was further tested in this study. The second objective was to quantify the impacts of the human-induced environmental changes, including the increase in air temperature, $CO_2$ concentration, and N deposition, on the C and N balances of the South Korean forests after the national forest rehabilitation plan (from 1973 to 2020). This quantification was performed by conducting a scenario analysis based on eight forcing scenarios, either including or not the increase in the air temperature, $CO_2$ concentration, and N deposition.

## 2. Materials and Methods

### 2.1. Study Forest

South Korea is in northeastern Asia. In 2020, the forests covered a total area of 6,287,000 ha [24]. The South Korean forests experience a temperate monsoon climate where approximately 70% of the mean annual precipitation is concentrated in summer (from June to August) [25]. The mean annual temperature and precipitation from 1973 to 2020 were 12.51 °C and 1315 mm, respectively [25]. The South Korean forests are remarkable for their young stand age, ranging from 30–40 years old in 2019 [24]. This is because the South Korean government restored the devastated forests through the national forest rehabilitation plan since 1973 [22]. This plan led the average stem volume to increase from 11.31 $m^3$ $ha^{-1}$ in 1973 to 161.45 $m^3$ $ha^{-1}$ in 2019 (Figure 1). In 2019, the primary forest types were as follows: (1) evergreen coniferous forests, (2) deciduous broad-leaved forests, and (3) mixed forests (combination of both evergreen coniferous and deciduous broad-leaved species), accounting for 36.9%, 32.0%, and 26.9% of the total area of the South Korean forests, respectively [24]. Primary soil types are inceptisol (65.1%), ultisol (13.5%), and entisol (10.8%) according to United States Department of Agriculture soil taxonomy [33].

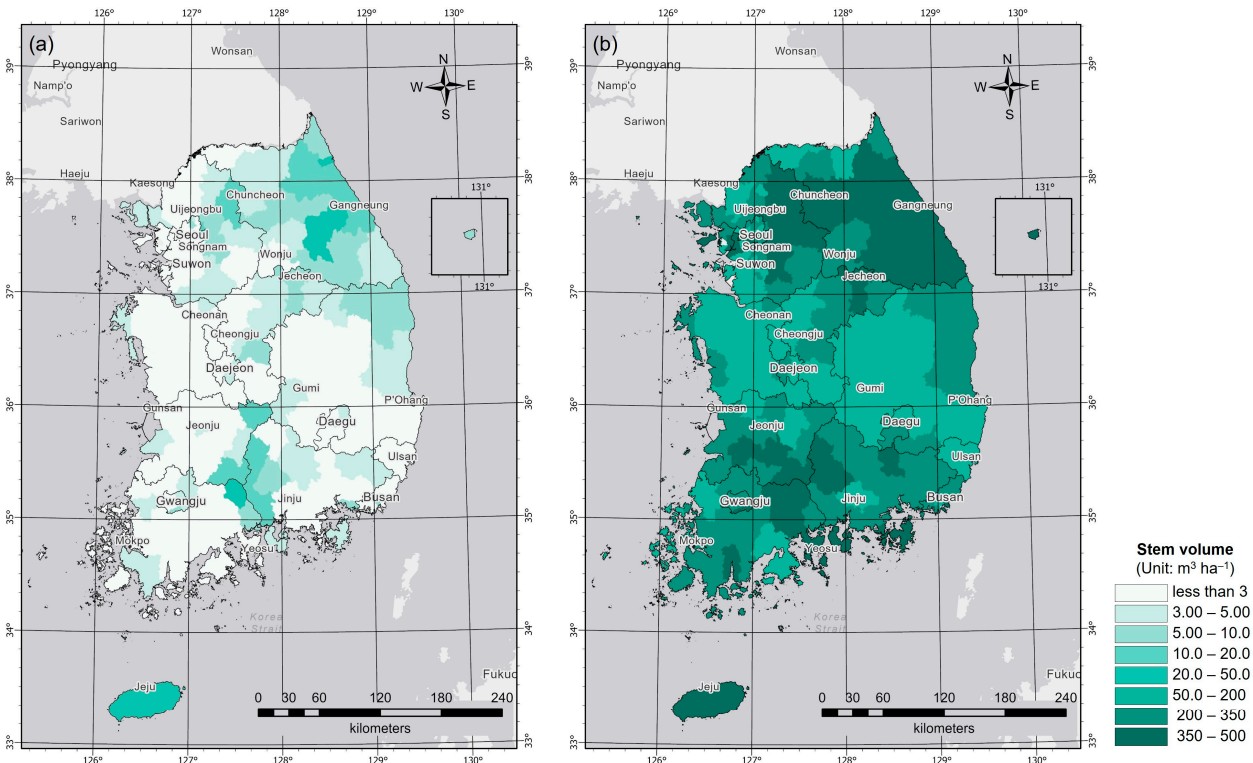

**Figure 1.** Forest maps of stem volume (**a**) before (in 1973) and (**b**) after the national forest rehabilitation plan (in 2019; data source: Korea Forest Service [24]; basemap sources: Esri, Redlands, CA, USA; HERE, Eindhoven, Netherlands; Garmin, Switzerland; © OpenStreetMap contributors; and the GIS User Community).

### 2.2. Model Description

We adopted the FBD-CAN in this study. The reliability and applicability of the FBD-CAN to the South Korean forests have been validated by a pilot study that simulated the C and N balances in a *Pinus densiflora* Siebold & Zucc. forest in central Korea [32] and a data-model fusion study, which estimated the mean C and N turnover times for the whole South Korean forests [19]. The spatial unit of model simulation is a stand where species, age, or environmental factors are distinguished from adjacent stands [34], and the temporal unit of model simulation can be daily, monthly, or annual. For the simulation, the FBD-CAN requires (1) spatially varying forcings, such as soil depth, forest type, initial stand age, and initial C and N stocks, and (2) spatiotemporally varying forcings, including

solar radiation, precipitation, air temperature, $CO_2$ concentration, and N deposition (both dry and wet deposition). The structure of the FBD-CAN comprises three main pools (tree biomass, primary DOM, and mineral soil) and multiple C and N fluxes (photosynthesis, C and N allocation, autotrophic respiration, turnover, heterotrophic respiration, N deposition, biological N fixation, N mineralization, N immobilization, tree N uptake, N resorption at turnover, and N output that includes N leaching and denitrification; Figure A1) [19]. The FBD-CAN uses two mechanisms to account for the interaction between C and N dynamics: (1) photosynthesis is constrained when inorganic N content in the mineral soil is lesser than the required N for the growth of tree biomass and N immobilization, and (2) N mineralization is limited when the C:N ratio of primary DOM or mineral soil is higher than the critical C:N ratio that determines whether N is mineralized or immobilized. Detailed information on the FBD-CAN is described in the previous studies [19,32].

*2.3. Database*

Soil depth data were extracted from the Forest Site and Soil Map provided by Korea Forest Service [35]. The forest type and initial stand age data were extracted from the Forest Type Map provided by the Korea Forest Service [36]. The initial C and N stocks for simulation start were generated using the spin-up process described in Lee et al. [28]. The solar radiation data were generated using the latitude and digital elevation model with the Area Solar Radiation tool in ArcGis Pro 2.6 [37]. The Korea Meteorological Administration provided the precipitation, air temperature, and $CO_2$ concentration data [25]. The N deposition data were extracted from the global N deposition map generated by the Coupled Model Intercomparison Project Phase 6, a collaborative framework to improve knowledge of climate change, using the Chemistry-Climate Model Initiative models with comprehensive stratosphere-troposphere chemistry [38,39]. Additional details on the type, source, and resolution of the forcings are provided in Table A1 in Appendix A.

We designed eight forcing scenarios with two levels (no change or ambient level versus observed or increased level) of each of the three environmental forcings of interest (air temperature, $CO_2$ concentration, and N deposition), but other forcings were identical in the eight scenarios (Table 1). The yearly average values during 1953–1972 were used to represent the ambient level of air temperature ($T_0$), $CO_2$ concentration ($CO_0$), and N deposition ($Nd_0$), whereas time-varying values from 1973 to 2020 were used to represent the increased level of air temperature ($T_1$), $CO_2$ concentration ($CO_1$), and N deposition ($Nd_1$; Figure A2). Among these eight scenarios, we defined the scenario that considers all the increase in the three environmental forcings ($T_1CO_1Nd_1$) as the "historical" scenario, and the scenario that does not consider all of them ($T_0CO_0Nd_0$) as the "control" scenario.

**Table 1.** Description of the eight forcing scenarios designed to assess the impacts of the national forest rehabilitation plan and human-induced environmental changes on the South Korean forests during 1973–2020.

| Scenarios | Level of Environmental Forcing | | |
| --- | --- | --- | --- |
| | **Air Temperature** | **$CO_2$ Concentration** | **N Deposition** |
| [1] $T_0CO_0Nd_0$ | Ambient | Ambient | Ambient |
| $T_0CO_0Nd_1$ | Ambient | Ambient | Increased |
| $T_0CO_1Nd_0$ | Ambient | Increased | Ambient |
| $T_0CO_1Nd_1$ | Ambient | Increased | Increased |
| $T_1CO_0Nd_0$ | Increased | Ambient | Ambient |
| $T_1CO_0Nd_1$ | Increased | Ambient | Increased |
| $T_1CO_1Nd_0$ | Increased | Increased | Ambient |
| [2] $T_1CO_1Nd_1$ | Increased | Increased | Increased |

Note: T, CO, and Nd stand for air temperature, $CO_2$ concentration, and N deposition, respectively. The subscript 0 and 1 indicate the ambient level and increased level, respectively. [1] a "control" scenario without increases in either of the three environmental forcings. [2] a "historical" scenario with increases in all three environmental forcings.

### 2.4. Model Simulation

The spatial simulation scale covered 6,056,400 ha of the South Korean forests, excluding the unstocked forest land and bamboo forest. This spatial simulation scale was subdivided into 60,564 spatial units (i.e., stands) of 100 ha each (Table A1). The temporal simulation scale was from 1973, when the national forest rehabilitation plan was launched, to 2020, and we set the temporal simulation unit as annual (Table A1).

For each of the eight scenarios (Table 1), we simulated the input, output, the balance of C and N, and the C and N stocks of tree biomass, primary DOM, and mineral soil of the South Korean forests during 1973–2020. The C input included the gross primary production of tree biomass, whereas the N input was calculated as the sum of N deposition and biological N fixation. The C output represented the sum of autotrophic respiration of tree biomass and heterotrophic respiration from primary DOM and mineral soil, whereas the N output included both N leaching and denitrification. The balance was computed as the difference between the input and output.

The simulation result from the "historical" scenario was used to assess the impacts of the national forest rehabilitation plan considering the three environmental forcings. The reliability of the simulation was verified by comparing the simulation results from the "historical" scenario with the observed values in National Forest Inventory (NFI) data, including the observed stem volume from 2006 to 2018 and the observed C and N stocks of dead wood, forest floor, and mineral soil in 2018.

The impacts of the human-induced environmental changes on the input, output, and balance of C and N during 1973–2020 were quantified by computing the relative difference between the simulation results under the seven forcing scenarios and those under the "control" scenario (Table 1). The C and N stocks of tree biomass, primary DOM, and mineral soil in 2020 were similarly assessed.

## 3. Results

### 3.1. Reliability of the Simulation

The stem volume simulated by the FBD-CAN was compared with the observed stem volume in NFI data from 2006 to 2018 (Figure 2a). The stem volume (mean $\pm$ standard deviation) simulated by the FBD-CAN was $179.5 \pm 106.7$ m$^3$ ha$^{-1}$, which was within the range of stem volume values ($185.9 \pm 124.2$ m$^3$ ha$^{-1}$) measured in the NFI. The increasing trend in the observed stem volume with a 7.0 Mg C ha$^{-1}$ year$^{-1}$ from 2006 to 2018 similarly appeared in the simulation by the FBD-CAN with a rate of 7.6 Mg C ha$^{-1}$ year$^{-1}$ during the same period.

The observed C and N stocks of dead wood, forest floor, and mineral soil in NFI data in 2018 were also compared with the simulation results of the FBD-CAN (Figure 2b–g). The simulated C stock of forest floor was $3.6 \pm 1.1$ Mg C ha$^{-1}$, which was similar to that observed in the NFI ($3.6 \pm 2.4$ Mg C ha$^{-1}$). The simulated C and N stocks of mineral soil were $38.5 \pm 11.9$ Mg C ha$^{-1}$ and $2.1 \pm 0.5$ Mg N ha$^{-1}$, respectively, which were similar to the observations in the NFI of $44.0 \pm 25.3$ Mg C ha$^{-1}$ and $2.8 \pm 2.8$ Mg N ha$^{-1}$ for the same C and N pools. Although the simulated C and N stocks in dead wood and N stock in mineral soil were mainly within the range of observed values from the NFI data, they could not cover the full range of variations in the observed values in the NFI data (Figure 2b,e,g).

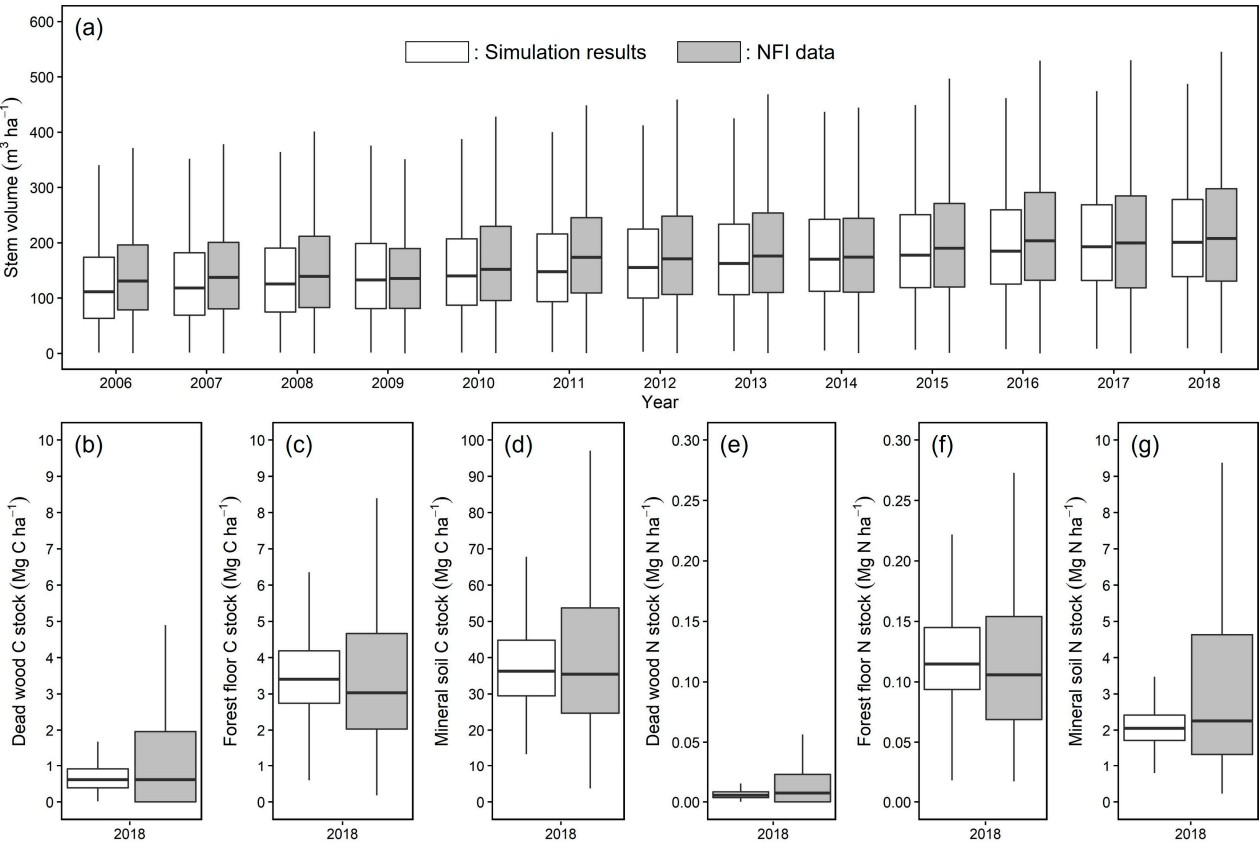

**Figure 2.** (**a**) Times-series comparison of the stem volume between the simulation results (white box plots) and the National Forest Inventory (NFI) data from 2006 to 2018 (gray box plots). Comparisons of (**b**) dead wood carbon (C) stock, (**c**) forest floor C stock, (**d**) mineral soil C stock, (**e**) dead wood nitrogen (N) stock, (**f**) forest floor N stock, and (**g**) mineral soil N stock between the simulation results (white box plots) and the NFI data in 2018 (gray box plots). The boxes' lower and upper limits indicate the 25% and 75% quartile, respectively. The bold line within the boxes denotes the median. The whiskers show the range of values from the minimum to maximum.

### 3.2. Carbon and Nitrogen Dynamics after the National Forest Rehabilitation Plan

Figure 3 shows the simulated dynamics of the mean annual C and N fluxes across the South Korean forests after the national forest rehabilitation plan. The C input and C output fluxes and their balance showed an increasing trend with time (Figure 3a). The C input flux increased from 6.2 Mg C ha$^{-1}$ year$^{-1}$ in 1973 to 25.1 Mg C ha$^{-1}$ year$^{-1}$ in 2020, whereas the C output flux increased from 5.9 Mg C ha$^{-1}$ year$^{-1}$ to 20.6 Mg C ha$^{-1}$ year$^{-1}$ between 1973 and 2020. The C balance in 1973 was only 0.3 Mg C ha$^{-1}$ year$^{-1}$, but it consistently increased to 4.1 Mg C ha$^{-1}$ year$^{-1}$ in 2010 owing to a faster increasing trend of the C input flux than that of the C output flux. Moreover, we observed that the increase in the C balance decreased in the 2010s. The increase in C balance from 2011 to 2020 was only 0.2 Mg C ha$^{-1}$ year$^{-1}$ per decade, whereas that from 1973 to 2010 was 1.1 Mg C ha$^{-1}$ year$^{-1}$ per decade.

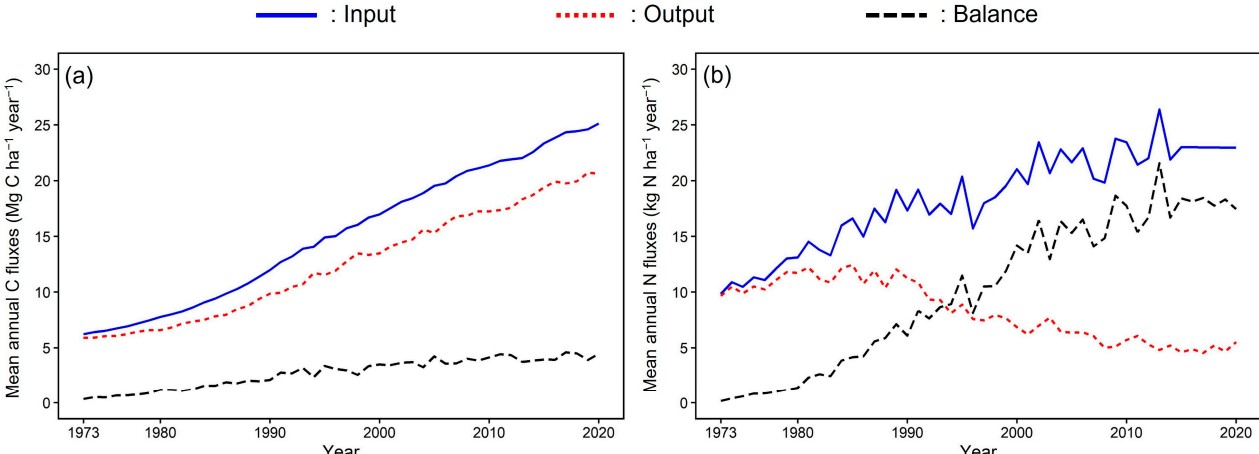

**Figure 3.** Dynamics of the simulated mean annual input, output, and balance of (**a**) C and (**b**) N across the South Korean forests from 1973 to 2020.

The N input flux increased from 9.8 kg N ha$^{-1}$ year$^{-1}$ in 1973 to 21.0 kg N ha$^{-1}$ year$^{-1}$ in 2000, but its increasing trend slowed after the 2000s (Figure 3b). In contrast, the N output flux slightly increased from 9.7 kg N ha$^{-1}$ year$^{-1}$ in 1973 to 12.4 kg N ha$^{-1}$ year$^{-1}$ in 1985, and then it continuously decreased over time to reach a value of 5.5 kg N ha$^{-1}$ year$^{-1}$ in 2020. The N balance increased from 0.2 kg N ha$^{-1}$ year$^{-1}$ in 1973 to 17.4 kg N ha$^{-1}$ year$^{-1}$ in 2020 because of the decrease in the N output flux after the mid-1980s and the increase in the N input flux from 1973 to the early 2000s.

The simulated dynamics of the mean C and N stocks across the South Korean forests after the national forest rehabilitation plan indicated that the total C stock increased from 40.1 Mg C ha$^{-1}$ in 1973 to 175.9 Mg C ha$^{-1}$ in 2020 at an average rate of 2.9 Mg C ha$^{-1}$ year$^{-1}$ (Figure 4a). The average annual increases in the C stocks of tree biomass, primary DOM, and mineral soil from 1973 to 2020 were 2.6, 0.1, and 0.2 Mg C ha$^{-1}$ year$^{-1}$, respectively. The mineral soil was the main C pool among the three C pools in 1973 storing 75% of the total C stock. However, the tree biomass became the major C pool in 2020 storing 72% of the total C stock because of the faster C accumulation of the tree biomass than that of the mineral soil. After the national forest rehabilitation plan, the whole South Korean forests stored 825 Tg C or 3025 Tg in CO$_2$ equivalent.

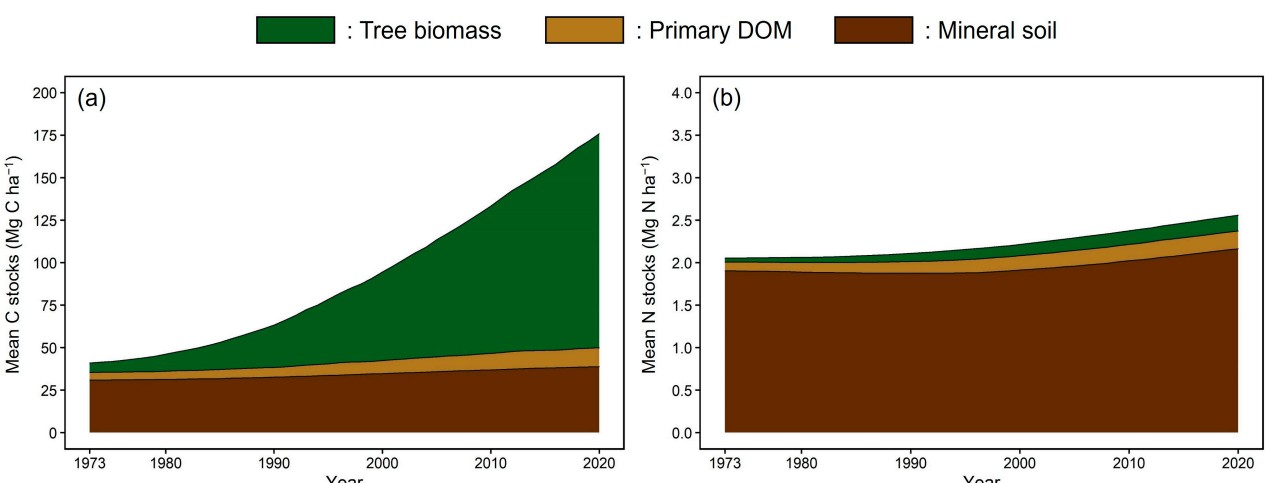

**Figure 4.** Dynamics of the simulated mean (**a**) C stocks and (**b**) N stocks of tree biomass, primary dead organic matter (DOM), and mineral soil across the South Korean forests from 1973 to 2020.

The total N stock increased from 2.1 Mg N ha$^{-1}$ in 1973 to 2.6 Mg N ha$^{-1}$ in 2020 at an average rate of 10.7 kg N ha$^{-1}$ year$^{-1}$ (Figure 4b). The average annual increases in

the N stocks of tree biomass, primary DOM, and mineral soil from 1973 to 2020 were 2.9, 2.3, and 5.5 kg N ha$^{-1}$ year$^{-1}$, respectively. In 1973, the mineral soil was the main N pool storing 93% of the total N stock, whereas the tree biomass and primary DOM stored only 2% and 5% of the total N stock, respectively. This trend was also observed in 2020; the tree biomass, primary DOM, and mineral soil stored 7%, 8%, and 85% of the total N stock, respectively. The whole South Korean forests stored 3.04 Tg N after the national forest rehabilitation plan.

### 3.3. Impacts of Human-Induced Environmental Changes on Carbon and Nitrogen Dynamics

During the simulation period from 1973 to 2020, using the increased level of air temperature as a forcing ($T_1CO_0Nd_0$) increased the mean average C input flux (+2.95%) and C output flux (+3.90%) across the South Korean forests (Table 2). However, the more significant increase in the C output flux decreased the C balance (−1.53%). The increase in $CO_2$ concentration ($T_0CO_1Nd_0$) increased the C balance by +2.64%. This increase in the C balance compensated for the decrease in the C balance due to an increase in air temperature, thereby increasing the C balance (+1.12%) when the increased level of both air temperature and $CO_2$ concentration were considered as forcings ($T_1CO_1Nd_0$). The increase in N deposition ($T_0CO_0Nd_1$) increased C input flux (+18.9%), C output flux (+17.3%), and C balance (+26.6%). The simultaneous increase in air temperature, $CO_2$ concentration, and N deposition ($T_1CO_1Nd_1$) increased C balance (+27.5%).

**Table 2.** The relative difference in the C and N fluxes during 1973–2020 across the South Korean forests as a result of an increased level of temperature, $CO_2$ concentration, and N deposition. Abbreviations of scenarios are found in Table 1.

| Scenarios | Relative Difference (%) | | | | | |
|---|---|---|---|---|---|---|
| | C Fluxes | | | N Fluxes | | |
| | Input | Output | Balance | Input | Output | Balance |
| $T_0CO_0Nd_0$ | – | – | – | – | – | – |
| $T_0CO_0Nd_1$ | +18.9 | +17.3 | +26.6 | +90.5 | +44.7 | +156 |
| $T_0CO_1Nd_0$ | +2.03 | +1.90 | +2.64 | 0 | −0.53 | +0.76 |
| $T_0CO_1Nd_1$ | +21.4 | +19.7 | +29.8 | +90.5 | +43.9 | +157 |
| $T_1CO_0Nd_0$ | +2.95 | +3.90 | −1.53 | 0 | +1.54 | −2.19 |
| $T_1CO_0Nd_1$ | +22.4 | +22.0 | +24.2 | +90.5 | +46.7 | +153 |
| $T_1CO_1Nd_0$ | +5.10 | +5.94 | +1.12 | 0 | +0.95 | −1.35 |
| $T_1CO_1Nd_1$ | +25.0 | +24.5 | +27.5 | +90.5 | +45.8 | +154 |

Note: Information about the abbreviations in scenarios was shown in Table 1.

Using the increased level of N deposition as a forcing ($T_0CO_0Nd_1$) for the simulation increased the mean average N input flux (+90.5%), N output flux (+44.7%), and N balance (+156%) across the South Korean forests from 1973 to 2020 compared with using the ambient level as a forcing (Table 2). The increase in air temperature ($T_1CO_0Nd_0$) increased the N output flux (+1.54%) and decreased the N balance (−2.19%). The increased $CO_2$ concentration ($T_0CO_1Nd_0$) caused a slight reduction in the N output flux (−0.53%) and an increase in the N balance (+0.76%). The simultaneous increase in air temperature, $CO_2$ concentration, and N deposition ($T_1CO_1Nd_1$) increased the N input flux (+90.5%), N output flux (+45.8%), and N balance (+154%).

Using the increased level of air temperature as a forcing for the simulation ($T_1CO_0Nd_0$) decreased the mean C stocks of tree biomass (−0.67 Mg C ha$^{-1}$), primary DOM (−0.55 Mg C ha$^{-1}$), and mineral soil (−0.67 Mg C ha$^{-1}$) in 2020 (Table 3). The increase in $CO_2$ concentration ($T_0CO_1Nd_0$) increased the C stocks in tree biomass (+2.36 Mg C ha$^{-1}$), primary DOM (+0.25 Mg C ha$^{-1}$), and mineral soil (+0.27 Mg C ha$^{-1}$). These increments offset the decrease in the C stocks due to the increase in air temperature, thereby increasing the C stock of tree biomass (+1.71 Mg C ha$^{-1}$) when both the increased air temperature and $CO_2$ concentration were considered forcings ($T_1CO_1Nd_0$). However, the increase in

$CO_2$ concentration could not fully offset the decrease in the C stocks of primary DOM and mineral soil due to the rise in air temperature ($T_1CO_1Nd_0$). The increase in N deposition ($T_0CO_0Nd_1$) increased the C stocks in the tree biomass (+23.4 Mg C ha$^{-1}$), primary DOM (+2.95 Mg C ha$^{-1}$), and mineral soil (+2.91 Mg C ha$^{-1}$). The simultaneous increase in air temperature, $CO_2$ concentration, and N deposition ($T_1CO_1Nd_1$) increased the C stocks of tree biomass (+25.1 Mg C ha$^{-1}$), primary DOM (+2.46 Mg C ha$^{--1}$), and mineral soil (+2.43 Mg C ha$^{-1}$). These results show that the human-induced increases in air temperature, $CO_2$ concentration, and N deposition contributed $-11.46$, 17.44, and 177.21 Tg C to the stored C stock of 825 Tg C in the South Korean forests, respectively.

**Table 3.** The difference in the mean C and N stocks in 2020 across the South Korean forests as a result of increased level of temperature, $CO_2$ concentration, and N deposition. Abbreviations of scenarios are found in Table 1.

| Scenarios | Difference | | | | | |
| --- | --- | --- | --- | --- | --- | --- |
| | C Stocks (Mg C ha$^{-1}$) | | | N Stocks (kg N ha$^{-1}$) | | |
| | Tree Biomass | Primary DOM | Mineral Soil | Tree Biomass | Primary DOM | Mineral Soil |
| $T_0CO_0Nd_0$ | – | – | – | – | – | – |
| $T_0CO_0Nd_1$ | +23.4 | +2.95 | +2.91 | +62.3 | +75.1 | +161 |
| $T_0CO_1Nd_0$ | +2.36 | +0.25 | +0.27 | $-1.97$ | $-0.43$ | +3.87 |
| $T_0CO_1Nd_1$ | +26.3 | +3.28 | +3.25 | +58.8 | +74.8 | +167 |
| $T_1CO_0Nd_0$ | $-0.67$ | $-0.55$ | $-0.67$ | +6.64 | $-6.17$ | $-4.66$ |
| $T_1CO_0Nd_1$ | +22.1 | +2.15 | +2.10 | +71.7 | +64.8 | +156 |
| $T_1CO_1Nd_0$ | +1.71 | $-0.31$ | $-0.41$ | +4.64 | $-6.50$ | $-0.71$ |
| $T_1CO_1Nd_1$ | +25.1 | +2.46 | +2.43 | +68.3 | +64.6 | +162 |

Note: Information about the abbreviations in scenarios was shown in Table 1.

Using the increased level of the air temperature as a forcing for the simulation ($T_1CO_0Nd_0$) increased the N stock in tree biomass (+6.64 kg N ha$^{-1}$) but decreased the N stocks in primary DOM ($-6.17$ kg N ha$^{-1}$) and mineral soil ($-4.66$ kg N ha$^{-1}$) in 2020 (Table 3). The increase in $CO_2$ concentration ($T_0CO_1Nd_0$) decreased N stocks in tree biomass ($-1.97$ kg N ha$^{-1}$) and primary DOM ($-0.43$ kg N ha$^{-1}$), whereas it increased the N stock in mineral soil (+3.87 kg N ha$^{-1}$). The increase in the N deposition ($T_0CO_0Nd_1$) resulted in an increase in tree biomass (+62.3 kg N ha$^{-1}$), primary DOM (+75.1 kg N ha$^{-1}$), and mineral soil (+161 kg N ha$^{-1}$). The simultaneous increase in air temperature, $CO_2$ concentration, and N deposition ($T_1CO_1Nd_1$) increased the N stocks of tree biomass (+68.3 kg N ha$^{-1}$), primary DOM (+64.6 kg N ha$^{-1}$), and mineral soil (+162 kg N ha$^{-1}$). These results indicate that the increase in air temperature, $CO_2$ concentration, and N deposition contributed $-25.38$, 8.90, and 1807.23 Mg N to the stored N stock of 3.04 Tg N in the South Korean forests, respectively.

## 4. Discussions

### 4.1. Carbon and Nitrogen Dynamics after the National Forest Rehabilitation Plan

The increases in the C and N stocks of tree biomass after the national forest rehabilitation plan reflect the fast growth of young forests, which has been widely acknowledged in previous studies [28–30]. We also found that C and N stocks of primary DOM and mineral soil slightly increased. However, a global meta-analysis showed that C stocks of soil for both organic and mineral horizons after afforestation in the temperate region remained unchanged, whereas those in the boreal, subtropical, and tropical regions increased [8]. This previous study also reported that the N stock of the mineral soil in temperate regions decreased because of the rapid N depletion in mineral soil during the first 5–10 years after afforestation [8]. Our findings on the increase in the C and N stocks of primary DOM and mineral soil might reflect the land-use history before the national forest rehabilitation plan was implemented. A global meta-analysis of the changes in the nutrients of mineral

soil showed that the C and N stocks of mineral soil slightly decreased after afforestation on grassland, moderately increased after afforestation on cropland, and substantially increased after afforestation on barren land [40]. This was because afforestation on barren land rapidly increases the vegetation cover [41], nutrient mineralization [42], and soil C stability by forming aggregates through a symbiotic association between plant roots and soil microbes [43]. Before the national forest rehabilitation plan was implemented, the South Korean forests were similar to a barren land because of devastation caused by forest exploitation under the Japanese occupation of Korea and the Korean War. Thus, the national forest rehabilitation plan for the devastated South Korean forests seemed to be more effective in the storage of the C and N of primary DOM and mineral soil because of the land use prior to the afforestation.

The responses of N dynamics to increasing N input can provide information on the N status of forests [44]. The NITRogen saturation EXperiments (NITREX) study showed that the increase in N input may lead to N-saturated forests, where the N availability exceeds the capacity of trees and soil microbes to accumulate N [45]. They also showed that N output increased as the forests reached the N-saturated state. However, in this study, N output decreased after the mid-1980s, whereas N input increased until the early 2000s. This is because the N output was limited by both the increase in N requirement of trees that had entered a fast-growing period after the mid-1980s and by the increase in N requirement of soil microbes for immobilization with the increase in the litter C input from tree biomass to primary DOM and mineral soil. Our findings on the opposite changing trend of N input and output with time might show that the increase in N input after the national forest rehabilitation plan was not sufficient to put the South Korean forests into N-saturated stage. It is more likely that the South Korean forests during 1973–2020 were still N-limited rather than N-saturated. This is supported by a review study that reported N-limited forests usually have less N output because of intense competition between trees and soil microbes for available N compared with the N-saturated forests [16]. The NITREX study also showed that increase in the N input did not accompany the increase in the N output, particularly in the N-limited forests [45]. Moreover, considering that the South Korean forests were barren before the national forest rehabilitation plan was implemented, they might have experienced severe N limitation [42].

### 4.2. Impacts of Human-Induced Environmental Changes on Carbon and Nitrogen Dynamics

Many findings in this study were consistent with previous studies on the responses of forest C dynamics to human-induced environmental changes. Over the last decades, researchers have concluded that photosynthetic C gain is stimulated by the increase in air temperature and $CO_2$ concentration, although there are saturation points for these two responses [46,47]. On the other hand, respirational C loss is assumed to increase exponentially with increased air temperature [48,49]. These results suggest that C dynamics of forests provide negative feedback to increasing air temperature and $CO_2$ concentration until air temperature exceeds the threshold where the stimulation effect of temperature on C loss exceeds the fertilization effect of $CO_2$ on the C gain [46]. The FBD-CAN, similar to other biogeochemical models, adopted this principle [19,50]. Our findings showed that the increase in air temperature and $CO_2$ concentration increased C balance of the South Korean forests due to increased tree biomass C stock. This might indicate that the increase in the air temperature over the past five decades in South Korea has not yet passed the temperature threshold at the whole ecosystem scale.

However, further increases in air temperature might pass this threshold for two reasons. First, the South Korean forests are aging. Younger trees usually show higher sensitivity to the $CO_2$ fertilization effect than mature trees, potentially owing to the decrease in the C use efficiency and increase in nutrient limitation with increasing stand age [51–53]. If the $CO_2$ fertilization effect decreases with the aging of the South Korean forests, even a slight temperature increase can negatively affect C balance. Second, in our simulation, the mean C stocks in 2020 of primary DOM and mineral soil decreased because of the

stimulated decomposition rate when the increased level of air temperature and $CO_2$ concentration is considered ($T_1CO_1Nd_0$). This indicates that the increased litter C input to the primary DOM and mineral soil because of the $CO_2$ fertilization effect on tree biomass could not fully offset the increased C output from the primary DOM and mineral soil due to stimulated decomposition with the increase in air temperature, similar to that described in a previous literature review [54]. This observation suggests that the increase in the air temperature over the last five decades had already exceeded the temperature threshold for the primary DOM and mineral soil when the increase in N deposition was not considered. Thus, a slight rise in air temperature in the future can exceed the temperature threshold and significantly reduce the C balance, considering that the decomposition rate increases exponentially with rising temperature [49] and that the $CO_2$ fertilization effect on the litter C input would decrease with the stand age [52].

It has been reported that other environmental factors, beyond $CO_2$ concentration and air temperature, can modify or even dominate the C balance of global forests because the interaction between forests and environmental factors varies with space and time [46]. There is empirical evidence on the interaction between the N deposition and C balance in forests [14,55,56]. For example, additional N deposition increased the foliar N concentration with a strong, positive effect on the C gain of trees, particularly in young and N-limited forests [31,54]. Moreover, increased C gain of trees owing to increased N deposition enhances the litter C input to the forest floor and soil, thereby increasing the C stock below the ground [57].

Accordingly, our simulation results for the effect of increased N deposition were consistent with the findings of previous studies. Notably, the response of the C balance to the increased N deposition dominated the responses to the increased air temperature and $CO_2$ concentration. This dominant effect of the increased N deposition on the C balance might be attributed to the young age of the South Korean forests because the sensitivity of the C balance to the increased N deposition is greater in young forests than in mature forests [58]. Additionally, before the national forest rehabilitation plan was implemented, the South Korean forests were barren land where severe N-limitation was expected to prevail, which can magnify the positive impact of additional N deposition on the C balance while limiting the positive effect of the $CO_2$ fertilization [59].

However, the surplus C gain of the South Korean forests owing to increased N deposition may not be sustained over time because the South Korean forests are transitioning to the mature forest stage, where the sensitivity of trees to the increased N deposition decreases. Moreover, N deposition in South Korea may not increase in the future, considering that the increase in N deposition had stopped after the 2010s with the efforts of the government to restrain N emissions from agricultural fertilization, transportation, and industry [27]. Therefore, if the N deposition of South Korea decreases, similar to that in other developed countries such as the United States and Europe, this could slow down tree growth and C sequestration once previously accumulated available N in mineral soil is depleted [60–62].

Our findings on the responses of N dynamics to human-induced environmental changes were consistent with those of previous studies [16,63,64]. The increase in N output with an increase in air temperature is owing to the stimulated decomposition and net N mineralization in primary DOM and mineral soil [65,66]. This stimulated net N mineralization with the increased air temperature increased N uptake and N stock of tree biomass, as also reported in a temperature manipulation study [66]. Meanwhile, N stock of mineral soil increased with the increase in $CO_2$ concentration, whereas both N output and N stock of tree biomass decreased. This is because of the increase in N uptake of soil microbes for N immobilization with enhanced litter C production due to the $CO_2$ fertilization effect, limiting the N uptake by tree biomass and N output through leaching and denitrification [67,68], which often occurs in N-limited forests [51].

The N input almost doubled with the increase in N deposition after the national forest rehabilitation plan in 1973; however, the rate of increase in N output was half that of the N

input. This indicates that approximately half of the increased N input has been incorporated into the internal N cycle without leaving the forest ecosystem [69,70]. Considering that the turnover time of N in the South Korean forests is approximately 450 years [19], the effect of increased N deposition on N dynamics is expected to persist in the near future, even without additional increases in N deposition. However, the impacts of increased air temperature and $CO_2$ concentration on N dynamics were marginal compared with that of increased N deposition because of the direct effects of N deposition on the N cycle, including tree N uptake, N leaching, and N immobilization.

### 4.3. Implications of the Findings in Managing South Korean Forests

The amount of $CO_2$ that has been stored in the South Korean forests after the national forest rehabilitation plan (3025 Tg in $CO_2$ eq in 2020) is approximately four times the amount of $CO_2$ emitted by the whole of South Korea (728 Tg in $CO_2$ eq in 2018) [71]. This shows that the national forest rehabilitation plan has been successful in terms of reducing the atmospheric $CO_2$ concentration.

However, the role of the South Korean forests in mitigating climate change may not continue without proper forest managements. This is because the $CO_2$ absorption capacity of a forest decreases over time as the forest ages [72–75]. Accordingly, our results on the dynamics of C balance over time showed that the South Korean forests have served as C sinks from 1973 to 2020, but the increasing trend in C balance stopped in the 2010s, likely indicating the beginning of the transition from the stage of young forests to mature forests [76]. These findings indicate that the $CO_2$ uptake capacity in the South Korean forests may decrease gradually in the near future.

Moreover, the positive impacts of increased $CO_2$ concentration and N deposition on the C balance would decrease with the aging of forests because of the lower sensitivity of mature forests to the fertilization effects of $CO_2$ concentration and N deposition than that of young forests [52,58]. Thus, to sustain the C sequestration role of the South Korean forests, forest management is needed to maintain the youth and productivity of the South Korean forests. Forest management activities, such as thinning that improves the tree growth [77], selective-harvesting that converts the C stock in tree biomass into a fixed and managed form of the harvested wood products [78], and afforestation after harvesting that transforms the mature forests into young forests with high net C uptake efficiencies [76], could be promising options for sustainable C sequestration by the South Korean forests in the future.

### 4.4. Limitations of the Simulation

The FBD-CAN used the mortality rate and stem volume to calculate the C and N stocks of dead wood. However, the model simulations did not fully cover the variation in the observed C and N stocks of deadwood, whereas the simulated values for stem volume were similar to the observed values. This indicates that the mortality rate in the FBD-CAN could be the cause of the discrepancy between the observed and simulated values for the C and N stocks of dead wood. Furthermore, although the FBD-CAN considers the species-specific mortality rate, this model does not account for the effect of other physical, chemical, and biological factors, including temperature, N availability, and stand density, on the mortality rate [69,79]. Moreover, the FBD-CAN does not consider the effect of natural events such as typhoons, fires, pests, and forest management on the mortality rate.

Similarly, the simulated N stock of mineral soil did not fully cover the range variation observed in the empirical data. This might be because the FBD-CAN does not fully account for soil microbial characteristics, such as species composition, life-cycle, and activity.

Our simulation might have uncertainties in the C and N dynamics responses to human-induced environmental changes because the FBD-CAN does not consider the acclimation of forests to changing environmental factors [32]. It has been reported that biogeochemical models that do not consider the acclimation of the responses to the increased air temperature and $CO_2$ concentration may overestimate the impacts of the interaction be-

tween the forest and atmosphere [80]. Notably, the maximum rate of Rubisco carboxylase activity ($V_{cmax}$), a photosynthetic parameter that determines the C gain of trees in the biogeochemical models, decreases under elevated $CO_2$ [81]; furthermore, dark respiration decreases under warmer temperature [82]. Thus, the FBD-CAN must consider the various factors affecting the mortality rate and soil microbial characteristics with the acclimation of responses to human-induced environmental changes to improve the reliability of the simulation.

## 5. Conclusions

This study used a biogeochemical modeling approach to assess the C and N dynamics of the South Korean forests after the national forest rehabilitation plan and quantify the contribution of human-induced environmental changes on the C and N dynamics. From 1973 to 2020, the C and N balances increased by several fold (21 and 87, respectively) owing to relatively low C and N output compared with the rapid integration of C and N to the young forests, indicating that the South Korean forests served as important C and N sinks. This led the whole South Korean forests of 6,056,400 ha to store 825 Tg C and 3.04 Tg N after the national forest rehabilitation plan. During 1973–2020, the human-induced increase in $CO_2$ concentration increased the C and N balances; however, the increase in air temperature decreased the C and N balances. We suggested forest management practices, such as thinning, selective-harvesting, and afforestation, to keep the South Korean forests young and productive under future air temperature, $CO_2$ concentration, and N deposition. However, further improvements in the FBD-CAN related to the mortality rate, microbial characteristics, and physiological acclimation to changing environmental factors are required for proper estimation of the impacts of human activities on the C and N balances in forests. Overall, this study provides insights on the benefits of forest rehabilitation for the C and N balances and may serve as a basis for future forest rehabilitation efforts. Moreover, our findings improve the understanding of the impacts of human activities on the C and N balances.

**Author Contributions:** H.-S.K., F.N., N.-J.N. and Y.-W.S. conceptualized the main idea; H.-S.K. conceived and designed methodology; H.-S.K. wrote and prepared original draft; H.-S.K., F.N., N.-J.N. and Y.-W.S. reviewed and edited the manuscript; H.-S.K. visualized research findings; Y.-W.S. acquired funding and supervised this study as a project administrator. All authors have read and agreed to the published version of the manuscript.

**Funding:** This study was carried out with support of 'R&D Program for Forest Science Technology (Project No. 2021363B10-2123-BD01)' provided by Korea Forest Service (Korea Forestry Promotion Institute) and with support of the Korea Agency for Infrastructure Technology Advancement grant funded by the Ministry of Land, Infrastructure and Transport (Grant 21UMRG-B158194-02).

**Institutional Review Board Statement:** Not applicable.

**Informed Consent Statement:** Not applicable.

**Data Availability Statement:** Detailed information and availability of data can be found in Table A1.

**Acknowledgments:** Florent Noulèkoun specifically acknowledges the support of the BK21 (Brain Korea 21 Program for Leading Universities and Students) FOUR program (Grant No. 4120200313708) and that of the Grant No. 2019R1I1A1A01064336, both funded by the National Research Foundation of Korea (NRF).

**Conflicts of Interest:** Authors declare no conflict of interest.

## Appendix A

**Table A1.** Detailed information of forcings for the simulation.

| Forcing | Forcing Type | Source | Resolution | |
| --- | --- | --- | --- | --- |
| | | | **Spatial** | **Temporal** |
| Soil depth | Spatial | Forest Site and Soil Map [35] | | |
| Forest type | Spatial | Forest Type Map [36] | 100 ha × 60,564 units | – |
| Initial stand age | Spatial | Forest Type Map [36] | | |
| Initial C and N stocks | Spatial | The spin-up process [28] | | |
| Solar radiation | Spatiotemporal | ArcGis Pro 2.6 [37] | | |
| Precipitation | Spatiotemporal | KMA [25] | | |
| Air temperature | Spatiotemporal | KMA [25] | 100 ha × 60,564 units | Annual, 1973–2020 |
| $CO_2$ concentration | Spatiotemporal | KMA [25] | | |
| N deposition | Spatiotemporal | CCMI [39] | | |

Note: C and N stand for carbon and nitrogen, respectively. KMA is Korea Meteorological Administration. CCMI is Chemistry-Climate Model Initiative.

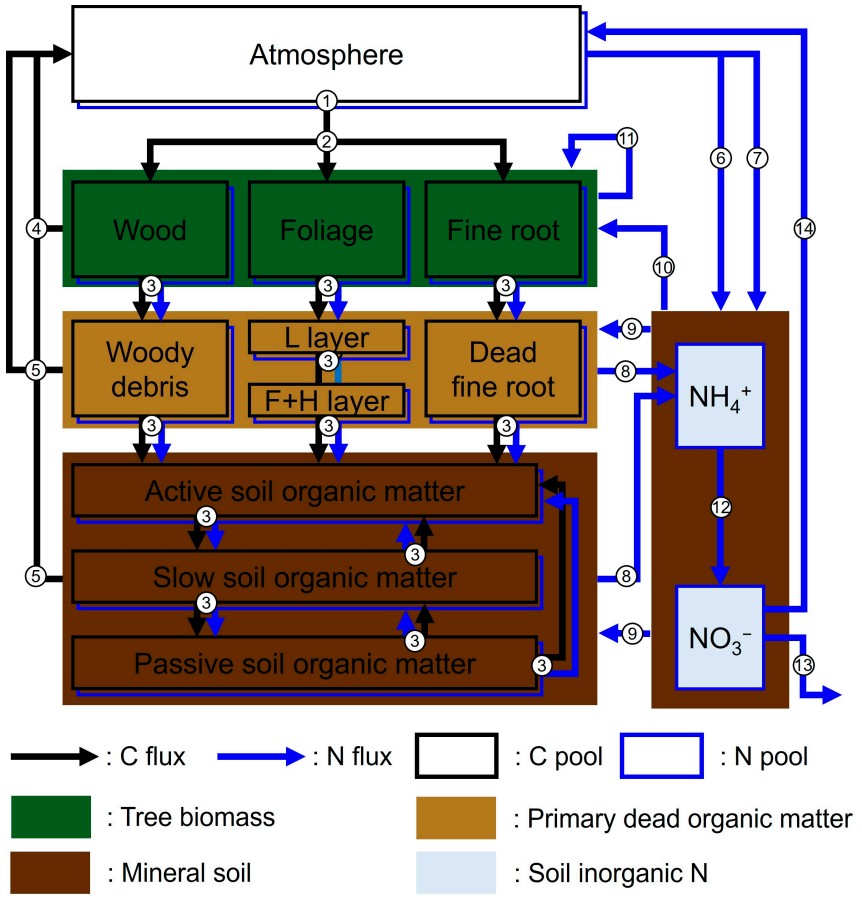

**Figure A1.** The structure of the Forest Biomass and Dead organic matter Carbon and Nitrogen (derived from Kim et al. [19]). ① Photosynthesis; ② C allocation; ③ Turnover; ④ Autotrophic respiration; ⑤ Heterotrophic respiration; ⑥ N deposition; ⑦ Biological N fixation; ⑧ N mineralization; ⑨ N immobilization; ⑩ Tree N uptake; ⑪ N resorption at turnover of foliage and fine root; ⑫ Nitrification; ⑬ N leaching; and ⑭ Denitrification). L and F+H layers denote the litter layer and the combined layer of fermentation and humus layers, respectively.

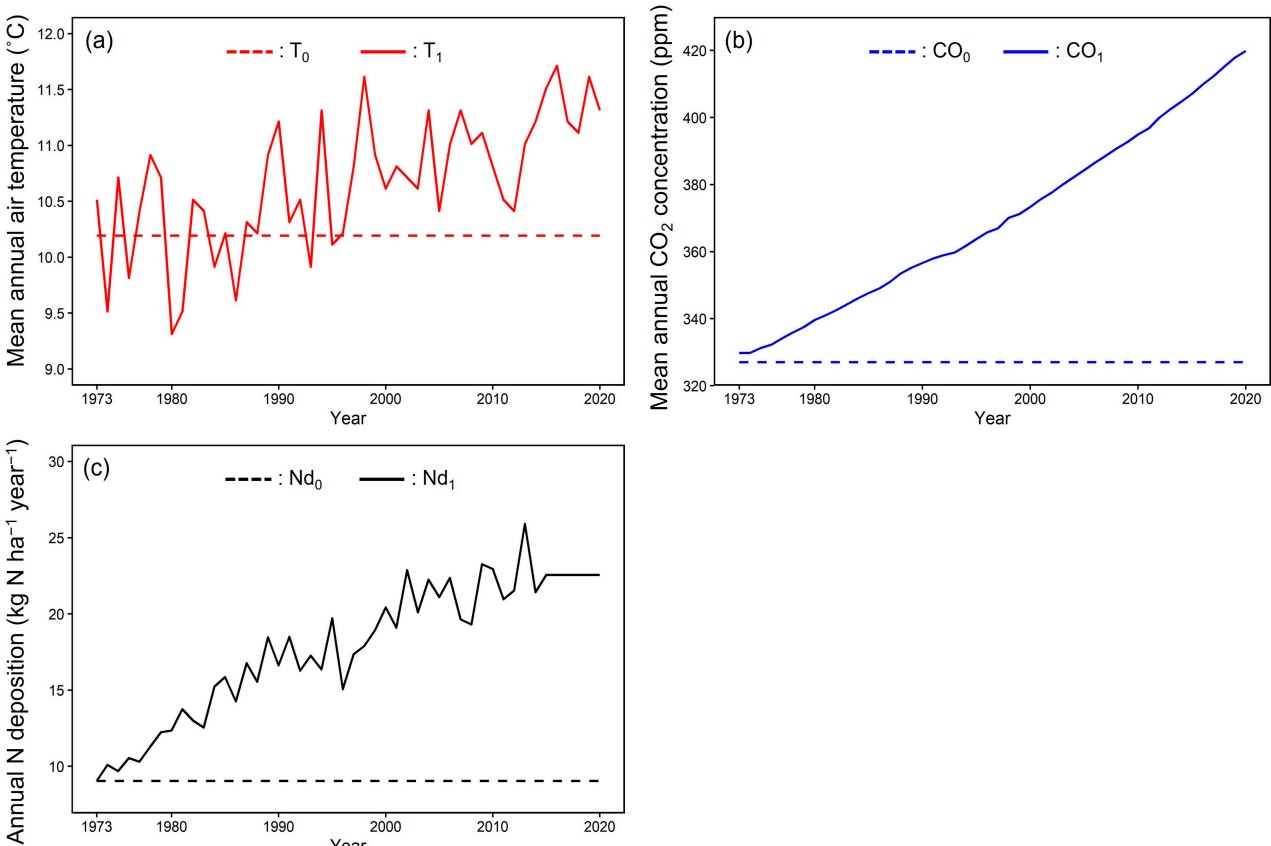

**Figure A2.** Environmental forcings of (**a**) air temperature, (**b**) $CO_2$ concentration, and (**c**) nitrogen deposition from 1973 to 2020 for ambient level (dashed lines; $T_0$, $CO_0$, and $Nd_0$) and increased level (solid lines; $T_1$, $CO_1$, and $Nd_1$).

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
