# Peer review of "Impacts of the National Forest Rehabilitation Plan and Human-Induced Environmental Changes on the Carbon and Nitrogen Balances of the South Korean Forests"

_forests, doi:10.3390/f12091150_

Round 1
Reviewer 1 Report
This study is interested about changes on the carbon and nitrogen balances of the South Korean forests after national forest rehabilitation plan. However, manuscript can more interested if author(s) can provide detail information about database and method, and can add some interested maps and figures for more better understanding. There is also lack of information about human induced environmental changes factors or drivers in this manuscript.
Please make sure all dash, space, a hyphen, en dash, and capital words would be appropriate throughout the manuscript.
Please make sure the font size in all figures, tables, and text.
Please articulate the novelty and the significance of the paper in the abstract, discussion and conclusion sections.
Abstract
Line 13-14: sorry it is unclear to me. responses of forests ?
2. Materials and Methods
I want to suggest that author(s) can add forest map (forest and non-forest) of South Korean before and after national forest rehabilitation plan in study area section.
Database and methods should be separated.
Please add detail information of database such as year, resolution etc.
4. Discussion
Discussion section nicely linked with international studies. However, there is still lack of information for South Korean perceptive. Author(s) should discuss how and why changes on the carbon and nitrogen balances of the South Korean forests before and after national forest rehabilitation plan influenced/impact by human activities (specific factors or drivers) and vice versa.
Reviewer 2 Report
This paper address the important issue of the impacts of human land-use on the carbon and nitrogen balance of forests, and the extent to which these impacts can be mitigated by re-forestation. The use of the FBD-CAN model to clarify these effects is novel, and the results of this study provide a solid basis on which to determine future policy on forest management.
I recommend this paper with minor edits (included in the attached PDF file). However, I do recommend an additional round of English language editing to correct any remaining errors.

Round 2
Reviewer 1 Report
I think, author(s) improved the manuscript based on Reviewer's comments.